# Multipole Graph Neural Operator for Parametric Partial Differential Equations

**Zongyi Li**
Caltech
zongyili@caltech.edu

**Nikola Kovachki**
Caltech
nkovachki@caltech.edu

**Kamyar Azizzadenesheli**
Purdue University
kamyar@purdue.edu

**Burigede Liu**
Caltech
bgl@caltech.edu

**Kaushik Bhattacharya**
Caltech
bhatta@caltech.edu

**Andrew Stuart**
Caltech
astuart@caltech.edu

**Anima Anandkumar**
Caltech
anima@caltech.edu

## Abstract

One of the main challenges in using deep learning-based methods for simulating physical systems and solving partial differential equations (PDEs) is formulating physics-based data in the desired structure for neural networks. Graph neural networks (GNNs) have gained popularity in this area since graphs offer a natural way of modeling particle interactions and provide a clear way of discretizing the continuum models. However, the graphs constructed for approximating such tasks usually ignore long-range interactions due to unfavorable scaling of the computational complexity with respect to the number of nodes. The errors due to these approximations scale with the discretization of the system, thereby not allowing for generalization under mesh-refinement. Inspired by the classical multipole methods, we propose a novel multi-level graph neural network framework that captures interaction at all ranges with only linear complexity. Our multi-level formulation is equivalent to recursively adding inducing points to the kernel matrix, unifying GNNs with multi-resolution matrix factorization of the kernel. Experiments confirm our multi-graph network learns discretization-invariant solution operators to PDEs and can be evaluated in linear time.

## 1 Introduction

A wide class of important scientific applications involve numerical approximation of parametric PDEs. There has been immense research efforts in formulating and solving the governing PDEs for a variety of physical and biological phenomena ranging from the quantum to the cosmic scale. While this endeavor has been successful in producing solutions to real-life problems, major challenges remain. Solving complex PDE systems such as those arising in climate modeling, turbulent flow of fluids, and plastic deformation of solid materials requires considerable time, computational resources, and domain expertise. Producing accurate, efficient, and automated data-driven approximation schemes has the potential to significantly accelerate the rate of innovation in these fields. Machine learning based methods enable this since they are much faster to evaluate and require only observational data to train, in stark contrast to traditional Galerkin methods [50] and classical reduced order models[40].

While deep learning approaches such as convolutional neural networks can be fast and powerful, they are usually restricted to a specific format or discretization. On the other hand, many problems can be naturally formulated on graphs. An emerging class of neural network architectures designed to operate on graph-structured data, Graph neural networks (GNNs), have gained popularity in this area. GNNs have seen numerous applications on tasks in imaging, natural language modeling, and the simulation of physical systems [47, 34, 31]. In the latter case, graphs are typically used to model particles systems (the nodes) and the their interactions (the edges). Recently, GNNs have been directly used to learn solutions to PDEs by constructing graphs on the physical domain [4], and it is was further shown that GNNs can learn mesh-invariant solution operators [32]. Since GNNs offer great flexibility in accurately representing solutions on any unstructured mesh, finding efficient algorithms is an important open problem.

The computational complexity of GNNs depends on the sparsity structure of the underlying graph, scaling with the number of edges which may grow quadratically with the number of nodes in fully connected regions [47]. Therefore, to make computations feasible, GNNs make approximations using nearest neighbor connection graphs which ignore long-range correlations. Such approximations are not suitable in the context of approximating solution operators of parametric PDEs since they will not generalize under refinement of the discretization, as we demonstrate in Section 4. However, using fully connected graphs quickly becomes computationally infeasible. Indeed evaluation of the kernel matrices outlined in Section 2.1 is only possible for coarse discretizations due to both memory and computational constraints. Throughout this work, we aim to develop approximation techniques that help alleviate this issue.

To efficiently capture long-range interaction, multi-scale methods such as the classical fast multipole methods (FMM) have been developed [22]. Based on the insight that long-range interaction are smooth, FMM decomposes the kernel matrix into different ranges and hierarchically imposes low-rank structures to the long-range components (hierarchical matrices)[11]. This decomposition can be viewed as a specific form of the multi-resolution matrix factorization of the kernel [29, 11]. However, the classical FMM requires nested grids as well as the explicit form of the PDEs. We generalize this idea to arbitrary graphs in the data-driven setting, so that the corresponding graph neural networks can learn discretization-invariant solution operators.

**Main contributions.**   Inspired by the fast multipole method (FMM), we propose a novel hierarchical, and multi-scale graph structure which, when deployed with GNNs, captures global properties of the PDE solution operator with a linear time-complexity [22, 48]. As shown in Figure 1, starting with a nearest neighbor graph, instead of directly adding edges to connect every pair of nodes, we add inducing points which help facilitate long-range communication. The inducing points may be thought of as forming a new subgraph which models long-range correlations. By adding a small amount of inducing nodes to the original graph, we make computation more efficient. Repeating this process yields a hierarchy of new subgraphs, modeling correlations at different length scales.

We show that message passing through the inducing points is equivalent to imposing a low-rank structure on the corresponding kernel matrix, and recursively adding inducing points leads to multi-resolution matrix factorization of the kernel [29, 11]. We propose the graph V-cycle algorithm (figure 1) inspired by FMM, so that message passing through the V-cycle directly computes the multi-resolution matrix factorization. We show that the computational complexity of our construction is linear in the number of nodes, achieving the desired efficiency, and we demonstrate the linear complexity and competitive performance through experiments on Darcy flow [44], a linear second-order elliptic equation, and Burgers' equation[43], which considered a stepping stone to Naiver-Stokes, is nonlinear, long-range correlated and more challenging. Our primary contributions are listed below.

- We develop the multipole graph kernel neural network (MGKN) that can capture long-range correlations in graph-based data with a linear time complexity in the nodes of the graph.

- We unify GNNs with multi-resolution matrix factorization through the V-cycle algorithm.

- We verify, analytically and numerically, the linear time complexity of the proposed methodology.

- We demonstrate numerically our method's ability to capture global information by learning mesh-invariant solution operators to the Darcy flow and Burgers' equations.

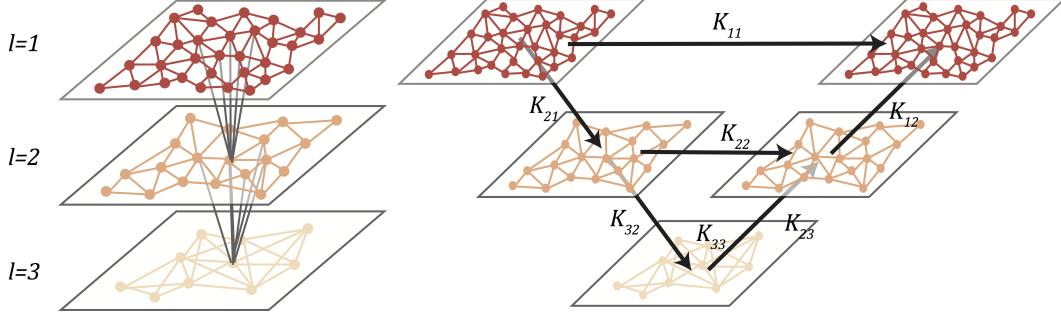

Figure 1: V-cycle

**Left**: the multi-level graph. **Right**: one V-cycle iteration for the multipole graph kernel network.

## 2 Operator Learning

We consider the problem of learning a mapping between two infinite-dimensional function spaces. Let $\mathcal{A}$ and $\mathcal{U}$ be separable Banach spaces and $\mathcal{F}^\dagger : \mathcal{A} \to \mathcal{U}$ be the target mapping. Suppose we have observation of $N$ pairs of functions $\{a_j, u_j\}_{j=1}^N$ where $a_j \sim \mu$ is i.i.d. sampled from a measure $\mu$ supported on $\mathcal{A}$ and $u_j = \mathcal{F}^\dagger(a_j)$, potentially with noise. The goal is to find an finite dimension approximation $\mathcal{F}_\theta : \mathcal{A} \to \mathcal{U}$, parametrized by $\theta \in \Theta$, such that

$$\mathcal{F}_\theta \approx \mathcal{F}^\dagger, \quad \mu - \text{a.e.}$$

We formulate this as an optimization problem where the objective $J(\theta)$ is the expectation of a loss functional $\ell : \mathcal{U} \times \mathcal{U} \to \mathbb{R}$. We consider the squared-error loss in an appropriate norm $\| \cdot \|_{\mathcal{U}}$ on $\mathcal{U}$ and discretize the expectation using the training data:

$$J(\theta) = \mathbb{E}_{a \sim \mu}[\ell(\mathcal{F}_\theta(a), \mathcal{F}^\dagger(a))] \approx \frac{1}{N} \sum_{j=1}^N \|\mathcal{F}_\theta(a_j) - \mathcal{F}^\dagger(a_j)\|_{\mathcal{U}}^2.$$

**Parametric PDEs.** An important example of the preceding problem is learning the solution operator of a parametric PDE. Consider $\mathcal{L}_a$ a differential operator depending on a *parameter* $a \in \mathcal{A}$ and the general PDE

$$(\mathcal{L}_a u)(x) = f(x), \qquad x \in D \tag{1}$$

for some bounded, open set $D \subset \mathbb{R}^d$, a fixed function $f$ living in an appropriate function space, and a boundary condition on $\partial D$. Assuming the PDE is well-posed, we may define the mapping of interest $\mathcal{F}^\dagger : \mathcal{A} \to \mathcal{U}$ as mapping $a$ to $u$ the solution of (1). For example, consider the second order elliptic operator $\mathcal{L}_a \cdot = -\text{div}(a\nabla \cdot)$. For a fixed $f \in L^2(D; \mathbb{R})$ and a zero Dirichlet boundary condition on $u$, equation (1) has a unique weak solution $u \in \mathcal{U} = H_0^1(D; \mathbb{R})$ for any parameter $a \in \mathcal{A} = L^\infty(D; \mathbb{R}^+) \cap L^2(D; \mathbb{R}^+)$. This example is prototypical of many scientific applications including hydrology [8] and elasticity [6].

Numerically, we assume access only to point-wise evaluations of the training pairs $(a_j, u_j)$. In particular, let $D_j = \{x_1, \ldots, x_n\} \subset D$ be a $n$-point discretization of the domain $D$ and assume we have observations $a_j|_{D_j}, u_j|_{D_j} \in \mathbb{R}^n$. Although the data given can be discretized in an arbitrary manner, our approximation of the operator $\mathcal{F}^\dagger$ can evaluate the solution at any point $x \in D$.

**Learning the Operator.** Learning the operator $\mathcal{F}^\dagger$ is typically much more challenging than finding the solution $u \in \mathcal{U}$ of a PDE for a single instance with the parameter $a \in \mathcal{A}$. Most existing methods, ranging from classical finite elements and finite differences to modern deep learning approaches such as physics-informed neural networks (PINNs) [38] aim at the latter task. And therefore they are computationally expensive when multiple evaluation is needed. This makes them impractical for applications such as inverse problem where we need to find the solutions for many different instances of the parameter. On the other hand, operator-based approaches directly approximate the operator and are therefore much cheaper and faster, offering tremendous computational savings when compared to traditional solvers.

- PDE solvers: solve one instance of the PDE at a time; require the explicit form of the PDE; have a speed-accuracy trade-off based on resolution: slow on fine grids but less accurate on coarse grids.

- Neural operator: learn a family of equations from data; don't need the explicit knowledge of the equation; much faster to evaluate than any classical method; no training needed for new equations; error rate is consistent with resolution.

## 2.1 Graph Kernel Network (GKN)

Suppose that $\mathcal{L}_a$ in (1) is uniformly elliptic then the Green's representation formula implies

$$u(x) = \int_D G_a(x,y)[f(y) + (\Gamma_a u)(y)]\, dy. \tag{2}$$

where $G_a$ is a Newtonian potential and $\Gamma_a$ is an operator defined by appropriate sums and compositions of the modified trace and co-normal derivative operators [41, Thm. 3.1.6]. We have turned the PDE (1) into the integral equation (2) which lends itself to an iterative approximation architecture.

**Kernel operator.**    Since $G_a$ is continuous for all points $x \neq y$, it is sensible to model the action of the integral operator in (2) by a neural network $\kappa_\phi$ with parameters $\phi$. To that end, define the operator $\mathcal{K}_a : \mathcal{U} \to \mathcal{U}$ as the action of the kernel $\kappa_\phi$ on $u$:

$$(\mathcal{K}_a u)(x) = \int_D \kappa_\phi(a(x), a(y), x, y) u(y)\, dy \tag{3}$$

where the kernel neural network $\kappa_\phi$ takes as inputs spatial locations $x, y$ as well as the values of the parameter $a(x), a(y)$. Since $\Gamma_a$ is itself an operator, its action cannot be fully accounted for by the kernel $\kappa_\phi$, we therefore add local parameters $W = wI$ to $\mathcal{K}_a$ and apply a non-linear activation $\sigma$, defining the iterative architecture $u^{(t)} = \sigma((W + \mathcal{K}_a)u^{(t-1)})$ for $t = 1, \ldots T$ with $u^{(0)} = a$. Since $\Gamma_a$ is local w.r.t $u$, we only need local parameters to capture its effect, while we expect its non-locality w.r.t. $a$ to manifest via the initial condition [41]. To increase expressiveness, we lift $u(x) \in \mathbb{R}$ to a higher dimensional representation $v(x) \in \mathbb{R}^{d_v}$ by a point-wise linear transformation $v_0 = Pu_0$, and update the representation

$$v^{(t)} = \sigma\big((W + \mathcal{K}_a)v^{(t-1)}\big), \qquad t = 1, \ldots, T \tag{4}$$

projecting it back $u^{(T)} = Qv^{(T)}$ at the last step. Hence the kernel is a mapping $\kappa_\phi : \mathbb{R}^{2(d+1)} \to \mathbb{R}^{d_v \times d_v}$ and $W \in \mathbb{R}^{d_v \times d_v}$. Note that, since our goal is to approximate the mapping $a \mapsto u$ with $f$ in (1) fixed, we do not need explicit dependence on $f$ in our architecture as it will remain constant for any new $a \in \mathcal{A}$.

For a specific discretization $D_j$, $a_j|_{D_j}, u_j|_{D_j} \in \mathbb{R}^n$ are $n$-dimensional vectors, and the evaluation of the kernel network can be viewed as a $n \times n$ matrix $K$, with its $x, y$ entry $(K)_{xy} = \kappa_\phi(a(x), a(y), x, y)$. Then the action of $\mathcal{K}_a$ becomes the matrix-vector multiplication $Ku$. For the lifted representation (4), for each $x \in D_j$, $v^{(t)}(x) \in \mathbb{R}^{d_v}$ and the output of the kernel $(K)_{xy} = \kappa_\phi(a(x), a(y), x, y)$ is a $d_v \times d_v$ matrix. Therefore $K$ becomes a fourth order tensor with shape $n \times n \times d_v \times d_v$.

**Kernel convolution on graphs.**    Since we assume a non-uniform discretization of $D$ that can differ for each data pair, computing with (4) cannot be implemented in a standard way. Graph neural networks offer a natural solution since message passing on graphs can be viewed as the integration (3). Since the message passing is computed locally, it avoids storing the full kernel matrix $K$. Given a discretization $D_j$, we can adaptively construct graphs on the domain $D$. The structure of the graph's adjacency matrix transfers to the kernel matrix $K$. We define the edge attributes $e(x, y) = (a(x), a(y), x, y) \in \mathbb{R}^{2(d+1)}$ and update the graph nodes following (4) which mimics the message passing neural network [21]:

$$v^{(t+1)}(x) = (\sigma(W + \mathcal{K}_a)v^{(t)})(x) \approx \sigma\Big(Wv^{(t)} + \frac{1}{|N(x)|}\sum_{y \in N(x)} \kappa_\phi\big(e(x,y)\big)v^{(t)}(y)\Big) \tag{5}$$

where $N(x)$ is the neighborhood of $x$, in this case, the entire discritized domain $D_j$.

**Domain of Integration.** Construction of fully connected graphs is memory intensive and can become computationally infeasible for fine discretizations i.e. when $|D_j|$ is large. To partially alleviate this, we can ignore the longest range kernel interactions as they have decayed the most and change the integration domain in (3) from $D$ to $B(x, r)$ for some fixed radius $r > 0$. This is equivalent to imposing a sparse structure on the kernel matrix $K$ so that only entries around the diagonal are non-zero and results in the complexity $O(n^2 r^d)$.

**Nyström approximation.** To further relieve computational complexity, we use Nyström approximation or the inducing points method by uniformly sampling $m < n$ nodes from the $n$ nodes discretization, which is to approximate the kernel matrix by a low-rank decomposition

$$K_{nn} \approx K_{nm} K_{mm} K_{mn} \tag{6}$$

where $K_{nn} = K$ is the original $n \times n$ kernel matrix and $K_{mm}$ is the $m \times m$ kernel matrix corresponding to the $m$ inducing points. $K_{nm}$ and $K_{mn}$ are transition matrices which could include restriction, prolongation, and interpolation. Nyström approximation further reduces the complexity to $O(m^2 r^d)$.

## 3 Multipole Graph Kernel Network (MGKN)

The fast multipole method (FMM) is a systematic approach of combining the aforementioned sparse and low-rank approximations while achieving linear complexity. The kernel matrix is decomposed into different ranges and a hierarchy of low-rank structures is imposed on the long-range components. We employ this idea to construct hierarchical, multi-scale graphs, without being constraint to particular forms of the kernel [48]. We elucidate the workings of the FMM through matrix factorization.

### 3.1 Decomposition of the kernel

The key to the fast multipole method's linear complexity lies in the subdivision of the kernel matrix according to the range of interaction, as shown in Figure 2:

$$K = K_1 + K_2 + \ldots + K_L \tag{7}$$

where $K_1$ corresponds to the shortest-range interaction, and $K_L$ corresponds to the longest-range interaction. While the uniform grids depicted in Figure 2 produce an orthogonal decomposition of the kernel, the decomposition may be generalized to arbitrary graphs by allowing overlap.

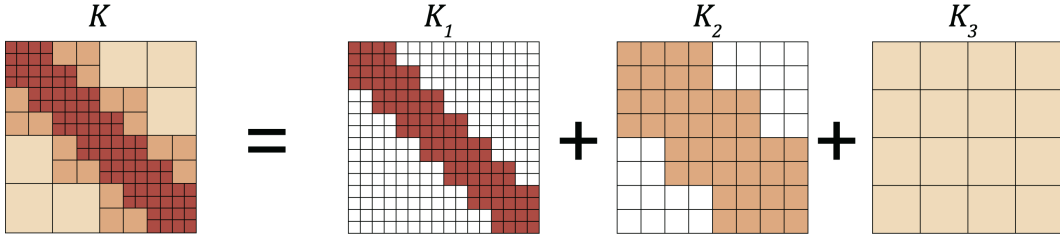

Figure 2: Hierarchical matrix decomposition
The kernel matrix $K$ is decomposed respect to ranges. $K_1$ corresponds to short-range interaction; it is sparse but high-rank. $K_3$ corresponds to long-range interaction; it is dense but low-rank.

### 3.2 Multi-scale graphs.

We construct $L$ graph levels, where the finest graph corresponds to the shortest-range interaction $K_1$, and the coarsest graph corresponds to the longest-range interaction $K_L$. In what follows, we will drop the time dependence from (4) and use the subscript $v_l$ to denote the representation at each level of the graph. Assuming the underlying graph is a uniform grid with resolution $s$ such that $s^d = n$, the $L$ multi-level graphs will be grids with resolution $s_l = s/2^{l-1}$, and consequentially $n_l = s_l^d = (s/2^{l-1})^d$ for $l = 1, \ldots, L$. In general, the underlying discretization can be arbitrary and we produce a hierarchy of $L$ graphs with a decreasing number of nodes $n_1, \ldots, n_L$.

The coarse graph representation can be understood as recursively applying an inducing points approximation: starting from a graph with $n_1 = n$ nodes, we impose inducing points of size $n_2, n_3, \ldots$ which all admit a low-rank kernel matrix decomposition of the form (6). The original $n \times n$ kernel matrix $K_l$ is represented by a much smaller $n_l \times n_l$ kernel matrix, denoted by $K_{l,l}$. As shown in Figure (2), $K_1$ is full-rank but very sparse while $K_L$ is dense but low-rank. Such structure can be achieved by applying equation (6) recursively to equation (7), leading to the multi-resolution matrix factorization [29]:

$$K \approx K_{1,1} + K_{1,2}K_{2,2}K_{2,1} + K_{1,2}K_{2,3}K_{3,3}K_{3,2}K_{2,1} + \cdots \tag{8}$$

where $K_{1,1} = K_1$ represents the shortest range, $K_{1,2}K_{2,2}K_{2,1} \approx K_2$, represents the second shortest range, etc. The center matrix $K_{l,l}$ is a $n_l \times n_l$ kernel matrix corresponding to the $l$-level of the graph described above. The long matrices $K_{l+1,l}, K_{l,l+1}$ are $n_{l+1} \times n_l$ and $n_{l+1} \times n_l$ transition matrices. We define them as moving the representation $v_l$ between different levels of the graph via an integral kernel that we learn. In general, $v^{(t)}(x) \in \mathbb{R}^{d_v}$ and the output of the kernel $(K_{l,l'})_{xy} = \kappa_\phi(a(x), a(y), x, y)$ is itself a $d_v \times d_v$ matrix, so all these matrices are again fourth-order tensors.

$$K_{l,l} : v_l \mapsto v_l = \int_{B(x,r_{l,l})} \kappa_{\phi_{l,l}}(a(x), a(y), x, y)v_l(y) \, dy \tag{9}$$

$$K_{l+1,l} : v_l \mapsto v_{l+1} = \int_{B(x,r_{l+1,l})} \kappa_{\phi_{l+1,l}}(a(x), a(y), x, y)v_l(y) \, dy \tag{10}$$

$$K_{l,l+1} : v_{l+1} \mapsto v_l = \int_{B(x,r_{l,l+1})} \kappa_{\phi_{l,l+1}}(a(x), a(y), x, y)v_{l+1}(y) \, dy \tag{11}$$

**Linear complexity.** The complexity of the algorithm is measured in terms of the sparsity of $K$ as this is what affects all computations. The sparsity represents the complexity of the convolution, and it is equivalent to the number of evaluations of the kernel network $\kappa$. Each matrix in the decomposition (7) is represented by the kernel matrix $K_{l,l}$ corresponding to the appropriate sub-graph. Since the number of non-zero entries of each row in these matrices is constant, we obtain that the computational complexity is $\sum_l O(n_l)$. By designing the sub-graphs so that $n_l$ decays fast enough, we can obtain linear complexity. For example, choose $n_l = O(n/2^l)$ then $\sum_l O(n_l) = \sum_l n/2^l = O(n)$. Combined with a Nyström approximation, we obtain $O(m)$ complexity.

### 3.3 V-cycle Algorithm

We present a V-cycle algorithm (not to confused with multigrid methods), see Figure 1, for efficiently computing (8). It consists of two steps: the **downward pass** and the **upward pass**. Denote the representation in downward pass and upward pass by $\check{v}$ and $\hat{v}$ respectively. In the downward step, the algorithm starts from the fine graph representation $\check{v}_1$ and updates it by applying a downward transition $\check{v}_{l+1} = K_{l+1,l}\check{v}_l$. In the upward step, the algorithm starts from the coarse presentation $\hat{v}_L$ and updates it by applying an upward transition and the center kernel matrix $\hat{v}_l = K_{l,l-1}\hat{v}_{l-1} + K_{l,l}\check{v}_l$. Notice that the one level downward and upward exactly computes $K_{1,1} + K_{1,2}K_{2,2}K_{2,1}$, and a full $L$-level v-cycle leads to the multi-resolution decomposition (8).

Employing (9)-(11), we use $L$ neural networks $\kappa_{\phi_{1,1}}, \ldots, \kappa_{\phi_{L,L}}$ to approximate the kernel $K_{l,l}$, and $2(L-1)$ neural networks $\kappa_{\phi_{1,2}}, \kappa_{\phi_{2,1}}, \ldots$ to approximate the transitions $K_{l+1,l}, K_{l,l+1}$. Following the iterative architecture (4), we also introduce the linear operator $W$, denoting it by $W_l$ for each corresponding resolution. Since it acts on a fixed resolution, we employ it only along with the kernel $K_{l,l}$ and not the transitions. At each time step $t = 0, \ldots, T-1$, we perform a full V-cycle:

**Downward Pass:**
$$\text{For } l = 1, \ldots, L: \qquad \check{v}_{l+1}^{(t+1)} = \sigma(\hat{v}_{l+1}^{(t)} + K_{l+1,l}\check{v}_l^{(t+1)}) \tag{12}$$

**Upward Pass:**
$$\text{For } l = L, \ldots, 1: \qquad \hat{v}_l^{(t+1)} = \sigma((W_l + K_{l,l})\check{v}_l^{(t+1)} + K_{l,l-1}\hat{v}_{l-1}^{(t+1)}). \tag{13}$$

We initialize as $v_1^{(0)} = Pu^{(0)} = Pa$ and output $u^{(T)} = Qv^{(T)} = Q\hat{v}_1^{(T)}$. The algorithm unifies multi-resolution matrix decomposition with iterative graph kernel networks. Combined with a Nyström approximation it leads to $O(m)$ computational complexity that can be implemented with message passing neural networks. Notice GKN is a specific case of V-cycle when $L = 1$.

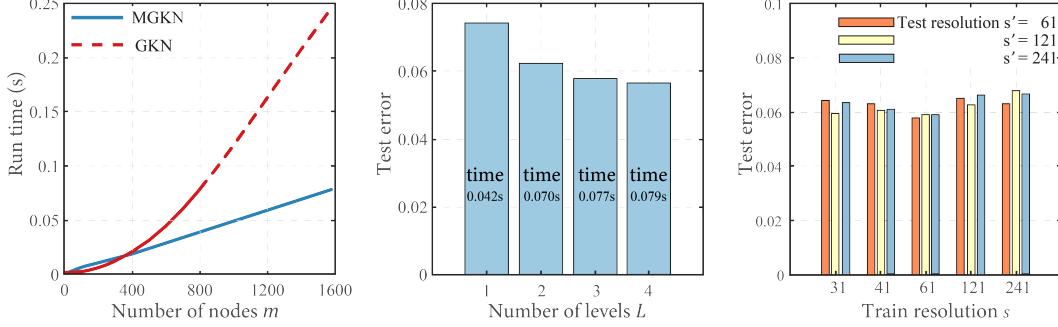

Figure 3: Properties of multipole graph kernel network (MGKN) on Darcy flow
**Left**: compared to GKN whose complexity scales quadratically with the number of nodes, MGKN has a linear complexity; **Mid**: Adding more levels reduces test error; **Right**: MGKN can be trained on a coarse resolution and perform well when tested on a fine resolution, showing invariance to discretization.

# 4    Experiments

We demonstrate the linear complexity and competitive performance of multipole graph kernel network through experiments on Darcy equation and Burgers equation.

## 4.1    Properties of the multipole graph kernel network

In this section, we show that MGKN has linear complexity and learns discretization invariant solutions by solving the steady-state of Darcy flow. In particular, we consider the 2-d PDE

$$
\begin{aligned}
-\nabla \cdot (a(x)\nabla u(x)) &= f(x) & x &\in (0,1)^2 \\
u(x) &= 0 & x &\in \partial(0,1)^2
\end{aligned}
\tag{14}
$$

and approximate the mapping $a \mapsto u$ which is non-linear despite the fact that (14) is a linear PDE. We model the coefficients $a$ as random piece-wise constant functions and generate data by solving (14) using a second-order finite difference scheme on a fine grid. Data of coarser resolutions are sub-sampled. See the supplements for further details. The code depends on Pytorch Geometric[19], also included in the supplements.

We use Nyström approximation by sampling $m_1, \ldots, m_L$ nodes for each level. When changing the number of levels, we fix coarsest level $m_L = 25, r_{L,L} = 2^{-1}$, and let $m_l = 25 \cdot 4^{L-l}, r_{l,l} = 2^{-(L-l)}$, and $r_{l,l+1} = r_{l+1,l} = 2^{-(L-l)+1/2}$. This set-up is one example that can obtain linear complexity. In general, any choice satisfying $\sum_l m_l^2 r_{l,l}^2 = O(m_1)$ also works. We set width $d_v = 64$, iteration $T = 5$ and kernel network $\kappa_{\phi_{l,l}}$ as a three-layer neural network with width $256/2^{(l-1)}$, so that coarser grids will have smaller kernel networks.

**1. Linear complexity:**    The left most plot in figure 3 shows that MGKN (blue line) achieves linear time complexity (the time to evaluate one equation) w.r.t. the number of nodes, while GKN (red line) has quadratic complexity (the solid line is interpolated; the dash line is extrapolated). Since the GPU memory used for backpropagation also scales with the number of edges, GKN is limited to $m \le 800$ on a single 11G-GPU while MGKN can scale to much higher resolutions . In other words, MGKN can be applicable for larger settings where GKN cannot.

**2. Comparing with single-graph:**    As shown in figure 3 (mid), adding multi-leveled graphs helps decrease the error. The MGKN depicted in blue bars starts from a fine sampling $L = 1; m = [1600]$, and adding subgraphs, $L = 2; m = [400, 1600]$, $L = 3; m = [100, 400, 1600]$, up to, $L = 4; m = [25, 100, 400, 1600]$. When $L = 1$, MGKN and GKN are equivalent. This experiment shows using multi-level graphs helps improve accuracy without increasing much of time-complexity.

**3. Generalization to resolution:**    The MGKN is discretization invariant, and therefore capable of super-resolution.We train with nodes sampled from a $s \times s$ resolution mesh and test on nodes sampled

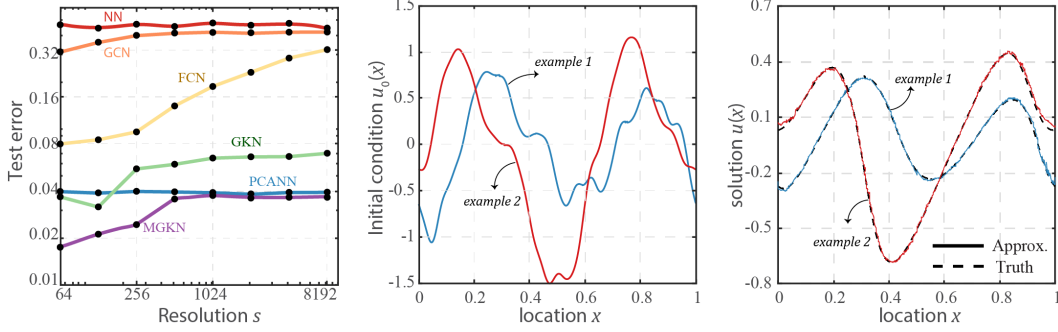

Figure 4: Comparsion with benchmarks on Burgers equation, with examples of inputs and outputs.
1-d Burgers equation with viscosity $\nu = 0.1$. **Left**: performance of different methods. MGKN has competitive
performance. **Mid**: input functions ($u_0$) of two examples. **Right**: corresponding outputs from MGKN of the
two examples, and their ground truth (dash line). The error is minimal on both examples..

from a $s' \times s'$ resolution mesh. As shown in on the right of figure 3, MGKN achieves similar testing
error on $s' = 61, 121, 241$, independently of the training discretization.

Notice traditional PDE solvers such as FEM and FDM approximate a single function and therefore
their error to the continuum decreases as resolution is increased. On the other hand, operator
approximation is independent of the ways its data is discretized as long as all relevant information is
resolved. Therefore, if we truly approximate an operator mapping, the error will be constant at any
resolution, which many CNN-based methods fail to preserve.

## 4.2 Comparison with benchmarks

We compare the accuracy of our methodology with other deep learning methods as well as reduced
order modeling techniques that are commonly used in practice. As a test bed, we use 2-d Darcy flow
(14) and the 1-d viscous Burger's equations:

$$\partial_t u(x,t) + \partial_x(u^2(x,t)/2) = \nu \partial_{xx} u(x,t), \qquad x \in (0, 2\pi), t \in (0, 1]$$
$$u(x,0) = u_0(x), \qquad x \in (0, 2\pi) \tag{15}$$

with periodic boundary conditions. We consider mapping the initial condition to the solution at
time one $u_0 \mapsto u(\cdot, 1)$. Burger's equation re-arranges low to mid range energies resulting in steep
discontinuities that are dampened proportionately to the size of the viscosity $\nu$. It acts as a simplified
model for the Navier-Stokes equation. We sample initial conditions as Gaussian random fields and
solve (15) via a split-step method on a fine mesh, sub-sampling other data as needed; two examples
of $u_0$ and $u(\cdot, 1)$ are shown in the middle and right of figure 4.

Figure 4 shows the relative test errors for a variety of methods on Burger's (15) (left) as a function of
the grid resolution. First notice that MGKN achieves a constant steady state test error, demonstrating
that it has learned the true infinite-dimensional solution operator irrespective of the discretization.
This is in contrast to the state-of-the-art fully convolution network (**FCN**) proposed in [49] which has
the property that what it learns is tied to a specific discretization. Indeed, we see that, in both cases,
the error increases with the resolution since standard convolution layer are parametrized locally and
therefore cannot capture the long-range correlations of the solution operator. Using linear spaces,
the (**PCA+NN**) method proposed in [10] utilizes deep learning to produce a fast, fully data-driven
reduced order model. The graph convolution network (**GCN**) method follows [4]'s architecture, with
naive nearest neighbor connection. It shows simple nearest-neighbor graph structures are insufficient.
The graph kernel network (**GKN**) employs an architecture similar to (4) but, without the multi-level
graph extension, it can be slow due the quadratic time complexity. For Burger's equation, when linear
spaces are no longer near-optimal, MGKN is the best performing method. This is a very encouraging
result since many challenging applied problem are not well approximated by linear spaces and can
therefore greatly benefit from non-linear approximation methods such as MGKN. For details on the
other methods, see the supplements.

The benchmark of the 2-d Darcy equation is given in Table 4, where MGKN again achieves a
competitive error rate. Notice in the 1-d Burgers' equation (Table 5), we restrict the transition matrices

$K_{l,l+1}, K_{l+1,l}$ to be restrictions and prolongations and thereby force the kernels $K_1, \ldots, K_L$ to be orthogonal. In the 2-d Darcy equation (Table 4), we use general convolutions (10, 11) as the transition, allowing overlap of these kernels. We observe the orthogonal decomposition of kernel tends to have better performance.

## 5 Related Works

**Deep learning approaches for PDEs:** There have been two primary approaches in the application of deep learning for the solution of PDEs. The first parametrizes the solution operator as a deep convolutional neural network (CNN) between finite-dimensional Euclidean spaces $\mathcal{F}_\theta : \mathbb{R}^n \to \mathbb{R}^n$ [23, 49, 3, 9, 45]. As demonstrated in Figure 4, such approaches are tied to a discritezation and cannot generalize. The second approach directly parameterizes the solution $u$ as a neural network $\mathcal{F}_\theta : D \to \mathbb{R}$ [16, 38, 7, 42, 27]. This approach is close to classical Galerkin methods and therefore suffers from the same issues w.r.t. the parametric dependence in the PDE. For any new parameter, an optimization problem must be solved which requires backpropagating through the differential operator $\mathcal{L}_a$ many times, making it too slow for many practical application. Only very few recent works have attempted to capture the infinite-dimensional solution operator of the PDE [4, 33, 32, 10, 36]. The current work advances this direction.

**GNN and non-sparse graphs:** A multitude of techniques such as graph convolution, edge convolution, attention, and graph pooling, have been developed for improving GNNs [28, 24, 21, 46, 35]. Because of GNNs' flexibility, they can be used to model convolution with different discretization and geometry [13, 14]. Most of them, however, have been designed for sparse graphs and become computationally infeasible as the number of edges grow. The work [5] proposes using a low-rank decomposition to address this issue. Works on multi-resolution graphs with a U-net like structure have also recently began to emerge and seen success in imaging, classification, and semi-supervised learning [39, 1, 2, 20, 30, 34, 31]. Our work ties together many of these ideas and provides a principled way of designing multi-scale GNNs. All of these works focus on build multi-scale structure on a given graph. Our method, on the other hand, studies how to construct randomized graphs on the spatial domain for physics and applied math problems. We carefully craft the multi-level graph that corresponds to multi-resolution decomposition of the kernel matrix.

**Multipole and multi-resolution methods:** The works [18, 17, 25] propose a similar multipole expansion for solving parametric PDEs on structured grids. Our work generalizes on this idea by allowing for arbitrary discretizations through the use of GNNs. Multi-resolution matrix factorizations have been proposed in [29, 26]. We employ such ideas to build our approximation architecture.

## 6 Conclusion

We introduced the multipole graph kernel network (MGKN), a graph-based algorithm able to capture correlations in data at any length scale with a linear time complexity. Our work ties together ideas from graph neural networks, multi-scale modeling, and matrix factorization. Using a kernel integration architecture, we validate our methodology by showing that it can learn mesh-invariant solutions operators to parametric PDEs. Ideas in this work are not tied to our particular applications and can be used provide significant speed-up for processing densely connected graph data.

## Acknowledgements

Z. Li gratefully acknowledges the financial support from the Kortschak Scholars Program. A. Anandkumar is supported in part by Bren endowed chair, LwLL grants, Beyond Limits, Raytheon, Microsoft, Google, Adobe faculty fellowships, and DE Logi grant. K. Bhattacharya, N. B. Kovachki, B. Liu and A. M. Stuart gratefully acknowledge the financial support of the Army Research Laboratory through the Cooperative Agreement Number W911NF-12-0022. Research was sponsored by the Army Research Laboratory and was accomplished under Cooperative Agreement Number W911NF-12-2-0022. The views and conclusions contained in this document are those of the authors and should not be interpreted as representing the official policies, either expressed or implied, of the Army Research Laboratory or the U.S. Government. The U.S. Government is authorized to reproduce and distribute reprints for Government purposes notwithstanding any copyright notation herein.

## Broader Impact

Many problems in science and engineering involve solving complex PDE systems repeatedly for different values of some parameters. Example arise in molecular dynamics, micro-mechanics, and turbulent flows. Often such systems exhibit multi-scale structure, requiring very fine discretizations in order to capture the phenomenon being modeled. As a consequence, traditional Galerkin methods are slow and inefficient, leading to tremendous amounts of resources being wasted on high performance computing clusters every day. Machine learning methods hold the key to revolutionizing many scientific disciplines by providing fast solvers that can work purely from data as accurate physical models may sometimes not be available. However traditional neural networks work between finite-dimensional spaces and can therefore only learn solutions tied to a specific discretizations. This is often an insurmountable limitation for practical applications and therefore the development of mesh-invariant neural networks is required. Graph neural networks offer a natural solution however their computational complexity can sometime render them ineffective. Our work solves this problem by proposing an algorithm with a linear time complexity that captures long-range correlations within the data and has potential applications far outside the scope of numerical solutions to PDEs. We bring together ideas from multi-scale modeling and multi-resolution decomposition to the graph neural network community.

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
