[Supplementary Material]

# Appendix

## 6.1 Table of Notations

Table 1: Table of notations

| Notation | Meaning |
|---|---|
| **PDE** | |
| $a \in \mathcal{A}$ | The input coefficient functions |
| $u \in \mathcal{U}$ | The target solution functions |
| $D \subset \mathbb{R}^d$ | The spatial domain for the PDE |
| $x \in D$ | The points in the the spatial domain |
| $\mathcal{F} : \mathcal{A} \rightarrow \mathcal{U}$ | The operator mapping the coefficients to the solutions |
| $\mu$ | Thee probability measure where $a_j$ sampled from. |
| **Graph Kernel Networks** | |
| $\mathcal{K}$ | The kernel integration operator |
| $\kappa : \mathbb{R}^{2(d+1)} \rightarrow \mathbb{R}^{n \times n}$ | The kernel maps $(x, y, a(x), a(y))$ to a $n \times n$ matrix |
| $\phi$ | The parameters of the kernel network $\kappa$ |
| $r$ | The radius of the kernel integration |
| $\sigma$ | The activation function |
| $t = 0, \ldots, T$ | The time steps |
| $v(x) \in \mathbb{R}^n$ | The neural network representation of $u(x)$ |
| $W$ | The linear operator |
| $P : u^{(0)} \mapsto v^{(0)}$ | The projection from the initialization to the representation |
| $Q : u^{(T)} \mapsto u^{(T)}$ | The projection from the representation to the solution |
| **Multi-graph** | |
| $n$ | Total number of nodes in the graph |
| $m$ | The number of sampled nodes (for the sampling method) |
| $l \in [1, \ldots, L]$ | The level, $l = 1$ is the finest level and $l = L$ the coarsest. |
| $r$ | The radius of the ball for the kernel integration |
| $K$ | The kernel matrix. |
| $K_{l,l}$ | The kernel matrix for level-$l$ subgraph. |
| $K_{l+1,l}$ | The transition matrix $K_{l+1,l} : v_l \mapsto v_{l+1}$. |
| $K_{l,l+1}$ | The transition matrix $K_{l,l+1} : v_{l+1} \mapsto v_l$. |
| $\check{v}$ | The representation $v$ in the downward pass. |
| $\hat{v}$ | The representation $v$ in the up pass. |
| **Hyperparameters** | |
| $N$ | The number of training pairs |
| $s$ | The underlying resolution of training points |
| $s'$ | The underlying resolution of testing points |

## 6.2 Experimental Details

**Data generation.** The steady-state Darcy flow equation used in Section 4.1 takes the form

$$\begin{aligned} -\nabla \cdot (a(x)\nabla u(x)) &= f(x) & x \in (0,1)^2 \\ u(x) &= 0 & x \in \partial(0,1)^2. \end{aligned} \tag{16}$$

The coefficients $a$ are generated according to $a \sim \mu$ where $\mu = \psi_{\#}\mathcal{N}(0, (-\Delta + 9I)^{-2})$ with a Neumann boundry condition on the operator $-\Delta + 9I$. The mapping $\psi : \mathbb{R} \rightarrow \mathbb{R}$ takes the value 12 on the positive part of the real line and 3 on the negative. Such constructions are prototypical models for many physical systems such as permeability in sub-surface flows and material microstructures in elasticity. Solutions $u$ are obtained by using a second-order finite difference scheme on $241 \times 241$ and $421 \times 421$ grids. Different resolutions are downsampled from this dataset.

The viscous Burgers' equation used in Section 4.2 takes the form

$$\begin{aligned} \partial_t u(x,t) + \partial_x(u^2(x,t)/2) &= \nu \partial_{xx} u(x,t), & x \in (0,2\pi), t \in (0,1] \\ u(x,0) &= u_0(x), & x \in (0,2\pi) \end{aligned} \tag{17}$$

with periodic boundary conditions. We consider mapping the initial condition to the solution at time one $u_0 \mapsto u(\cdot, 1)$. The initial condition is generated according to $u_0 \sim \mu$ where $\mu = \mathcal{N}(0, 625(-\Delta + 25I)^{-2})$ with periodic boundary conditions. We set the viscosity to $\nu = 0.1$ and solve the equation using a split step method where the heat equation part is solved exactly in Fourier space then the non-linear part is advanced, again in Fourier space, using a very fine forward Euler method. We solve on a spatial mesh with resolution $2^{13} = 8192$ and use this dataset to subsample other resolutions.

**Linear complexity and comparison with GKN.** The first two block-rows in table 2 correspond to results of MGKN on the Darcy flow problem when using a different number of subgraphs with their respective number of nodes given by the vector $m$. For example $m = [400, 100, 25]$ means a multi-graph of three levels with nodes $400, 100, 25$ respectively for each level. The third and fourth block-rows correspond to GKN, where, in the third block, we fix the domain of integration to be a ball with radius $r = 0.1$, and, in the fourth block, a ball with radius $r = 0.2$. The number $m$ corresponds to the number of nodes sampled in the Nyström approximation. All reported errors are relative $L^2$ errors. The training time corresponds to 1 epoch using $N = 100$ data pairs, while the testing time corresponds to evaluating the PDE for 100 new queries. The left and middle images of Figure 3 are constructed from this data.

Table 2: Darcy flow experimental results.

| | training error | testing error | training time | testing time |
|---|---|---|---|---|
| $m = [25]$ | 0.0181 | 0.1054 | $0.50s$ | $0.21s$ |
| $m = [100, 25]$ | 0.0189 | 0.0781 | $1.59s$ | $0.69s$ |
| $m = [400, 100, 25]$ | 0.0155 | 0.0669 | $5.24s$ | $1.90s$ |
| $m = [1600, 400, 100, 25]$ | 0.0165 | 0.0568 | $19.75s$ | $7.94s$ |
| $m = [1600]$ | 0.0679 | 0.0744 | $10.74s$ | $4.28s$ |
| $m = [1600, 400]$ | 0.0359 | 0.0624 | $17.78s$ | $7.04s$ |
| $m = [1600, 400, 100]$ | 0.0184 | 0.0578 | $19.50s$ | $7.70s$ |
| $m = [1600, 400, 100, 25]$ | 0.0165 | 0.0568 | $19.75s$ | $7.94s$ |
| $m = 25, r = 0.1$ | 0.0994 | 0.1139 | $0.45s$ | $0.16s$ |
| $m = 100, r = 0.1$ | 0.0824 | 0.1067 | $0.44s$ | $0.16s$ |
| $m = 400, r = 0.1$ | 0.0514 | 0.0769 | $1.76s$ | $0.64s$ |
| $m = 1600, r = 0.1$ | 0.0470 | 0.0588 | $24.81s$ | $8.96s$ |
| $m = 25, r = 0.2$ | 0.0728 | 0.1169 | $0.45s$ | $0.16s$ |
| $m = 100, r = 0.2$ | 0.0508 | 0.0803 | $0.59s$ | $0.19s$ |
| $m = 200, r = 0.2$ | 0.0438 | 0.0668 | $1.53s$ | $0.55s$ |
| $m = 400, r = 0.2$ | 0.0462 | 0.0601 | $5.56s$ | $2.13s$ |
| $m = 800, r = 0.2$ | 0.0397 | 0.0519 | $22.31s$ | $7.86s$ |

**Mesh invariance.** As shown in table 3, MGKN can be trained on data with resolution $s$ and be evaluated on with data with resolution $s'$. We train a MGKN model for each of the choices $s = 31, 41, 61, 121, 241$. The table demonstrates that we achieve consistently low error on any pair of train-test resolutions hence we learn an infinite-dimensional mapping that is resolution invariant. The right image in Figure 3 was generated using this data.

Table 3: Generalization of resolutions on sampled Grids

| **Resolutions** | $s' = 61$ | $s' = 121$ | $s' = 241$ |
|---|---|---|---|
| $s = 31$ | 0.0643 | 0.0594 | 0.0634 |
| $s = 41$ | 0.0630 | 0.0606 | 0.0609 |
| $s = 61$ | 0.0579 | 0.0590 | 0.0589 |
| $s = 121$ | 0.0650 | 0.0628 | 0.0664 |
| $s = 241$ | 0.0630 | 0.0680 | 0.0665 |

**Comparison with benchmarks.** Table 4 and 5 show the performance of different methods on Darcy flow and Burgers' equation respectively. The training size $N = 1000$; the testing size $N = 100$.

- **NN** is a simple point-wise feedforward neural network. It is mesh-free, but performs badly due to lack of neighbor information. NN represents the baseline of a local map.

- **GCN**, the graph convolution network, follows the architecture in [4], with naive nearest neighbor connection. Such nearest-neighbor graph structure has acceptable error for very coarse grid ($s = 16$). For common resolution $s = 64$, nearest-neighbor graph can only capture a near-local map, similar to NN. It shows simple nearest-neighbor graph structures are insufficient.

- **FCN** is the state of the art neural network method based on Fully Convolution Network [49]. It has a good performance for small grids $s = 61$. But fully convolution networks are mesh-dependent and therefore their error grows when moving to a larger grid.

- **PCA+NN** is an instantiation of the methodology proposed in [10]: using PCA as an autoencoder on both the input and output data and interpolating the latent spaces with a neural network. The method provably obtains mesh-independent error and can learn purely from data, however the solution can only be evaluated on the same mesh as the training data. Furthermore the method uses linear spaces, justifying its strong performance on diffusion dominated problems such as Darcy flow. This is further discussed below.

- **RBM** is the reduced basis method [15], a classical reduced order modeling technique that is ubiquitous in applications [37] . It approximates the solution operator within an linear class of basis functions, requiring data as well as the variational form the problem. Since the solution manifold of (14) exhibits fast decay of its Kolgomorov $n$-width, linear spaces are near optimal hence it is not surprising that RBM is the best performing method [12]. Compared to deep learning approaches, RMB is significantly slower as it requires numerical integration to form and then invert a linear system for every new parameter.

- **GKN** stands for graph kernel network with $r = 0.25$ and $m = 300$. It enjoys competitive performance against all other methods while being able to generalize to different mesh geometries. The drawback is its quadratic complexity, constrain GKN from a large radius. Therefore GKN has higher error rates on the Burgers equation where long-range correlation is not negligible.

- **MKGN** is our new proposed method. MKGN has slight higher error on Darcy flow where linear spaces are near-optimal. but for the harder Burger's equation, MGKN is the best performing method. This is a very encouraging result since many challenging applied problem are not well approximated by linear spaces and can therefore greatly benefit from non-linear approximation methods such as MGKN.

Table 4: Error of different methods on Darcy Flow

| **Networks** | $s = 85$ | $s = 141$ | $s = 211$ | $s = 421$ |
|---|---|---|---|---|
| NN | 0.1716 | 0.1716 | 0.1716 | 0.1716 |
| GCN | 0.1356 | 0.1414 | 0.1482 | 0.1579 |
| FCN | 0.0253 | 0.0493 | 0.0727 | 0.1097 |
| PCA+NN | 0.0299 | 0.0298 | 0.0298 | 0.0299 |
| RBM | 0.0244 | 0.0251 | 0.0255 | 0.0259 |
| GKN | 0.0346 | 0.0332 | 0.0342 | 0.0369 |
| MGKN | 0.0416 | 0.0428 | 0.0428 | 0.0420 |

Table 5: Error of different methods on Burgers' equation

| Networks | $s = 64$ | $s = 128$ | $s = 256$ | $s = 512$ | $s = 1024$ | $s = 2048$ | $s = 4096$ | $s = 8192$ |
|---|---|---|---|---|---|---|---|---|
| NN | 0.4677 | 0.4447 | 0.4714 | 0.4561 | 0.4803 | 0.4645 | 0.4779 | 0.4452 |
| GCN | 0.3135 | 0.3612 | 0.3999 | 0.4138 | 0.4176 | 0.4157 | 0.4191 | 0.4198 |
| FCN | 0.0802 | 0.0853 | 0.0958 | 0.1407 | 0.1877 | 0.2313 | 0.2855 | 0.3238 |
| PCA+NN | 0.0397 | 0.0389 | 0.0398 | 0.0395 | 0.0391 | 0.0383 | 0.0392 | 0.0393 |
| GKN | 0.0367 | 0.0316 | 0.0555 | 0.0594 | 0.0651 | 0.0663 | 0.0666 | 0.0699 |
| MGKN | 0.0174 | 0.0211 | 0.0243 | 0.0355 | 0.0374 | 0.0360 | 0.0364 | 0.0364 |