[Reviews · NeurIPS 2020]

Review 1

Summary and Contributions: I have read the author's feedback. I really appreciate the time you took to answer us, reviewers. Rest assured that I have devoted a substantial amount of time looking at your paper and response. Unfortunately, the authors explanations do not add much to what I had already understood, and do not fully settle my concerns. Examples follow. On a few "subjective" matters: I disagree with the authors' response on "technicality" and "graphs" and I strongly suggest they take my advice into consideration in a future revision. In particular, regarding the use of graphs, even if the authors use graph-based software, the explanation do not need them. Examples: I appreciate the motivating example with PDE with stochastic boundaries. However, the authors should have also reported the training time, and the time to generate the training data. In a nutshell, a few more numbers would have made all the difference. Also, the authors should have addressed the obvious possibility of using methods that can start from the solution for a given $a$ to quickly find the solution for $a'$ close to $a$. I think that being a bit more self-critical would have probably helped the paper as well. I appreciate the promise for the two new examples. The authors should have done such experiments. It would have made all the difference to give a few numbers about their outcome. ====================================== The authors consider the problem of solving a PDE of the kind L_a u(x) = f(x) where L_a is a differential operator that depends on a parameter $a$, a function, and an input function f. The authors consider this problem in the scenario where f is fix but we want to solve the PDE over varying "values" for the parameter $a$ very fast. The authors also assume that we have access to a training set of pairs $(a(x),u(x))$ that are examples of solutions to the PDE. To tackle this problem, the authors first express the solution of the PDE as the fixed point of an integral recurrence, and then approximate the integral operator in the recurrence with a neural network, thus expressing the successive applications of the integral recurrence as the output of the composition of several NNs. These NN are build such that the computation time of the overall map remains tractable. In particular, they express the integral operator as the composition and sum of simpler NNs, each representing operators at a given "scale", i.e. level of detail. The authors compare their method with existing NN based methods for the same problem, and argue that their method should better performance.

Strengths: The authors work is a combination of several existing ideas, as they point out. The general idea of replacing integral operators with NNs is interesting. The general idea of expressing this operator as a sum and composition of operators that are sparse/dense & low-rank/high-rank is also interesting.

Weaknesses: There are several important points that need to be addressed in the paper. First, there is a non-uniform level of detail and technicality through out the paper. The authors start by trying to be very formal, and specifying that functions come from "separable Banach spaces", but quickly drop this rigor and start being vague mentioning "appropriate function-space" and "well-posed", and in the end, the actual method that they propose does not need these details, and is often explained using concepts, like "graphs", that were never formally defined. I suggest using the following heuristic: If a concept is not needed to explain how you get to eq. (12) and (13), then do not mention it at all. This will save you space that you should then use to explain what actually matters in more detail. Second, although I appreciate the comparison with other NN-based methods, it would be important for the authors to illustrate with an example how traditional methods perform. I do understand that traditional methods require recalculations for each new parameter $a$. Nonetheless, the numerical examples studied are not too complex, and I suspect that standard PDE solvers could solve them really fast. It would be good to report these time-to-solution numbers, as well as the accuracy of these classical solvers. This is also relevant for the numerical section, for one to understand how exactly the training set was generated. I assume you used some classical solver to find the training examples, yes ? Also, what is the training time? Sorry, didn't check the Appendix (as a reviewer I don't have to), maybe it is there?! In any case, a few of these numbers should be in the main text. Are you familiar with the Matlab package Chebfun? It can use Chebyshev interpolation (instead of grid-type discretizations like you do at each level) to solve PDEs very accurately, and it is really easy to use. I would be very interested in seeing how well you do in comparison with these type of tools. They also allow the computation pre-compuation of inverse operators, so, with some tricks, it could even be that as you change $a$, you do not need to recompute much stuff. Third, some of the explanations need to be clarified. I mention a few now. 1) Fig. 1 is pretty but it is not very informative. It is too generic, and it is incomprehensible at the point where it is first mentioned. I suggest that you move it to later, to where you explain your V-cycle, Section 3.2, and then also had \hat{v} and \invertedhat{v} to the different parts of the diagram. Also, I recommend not pointing the arrows inwards as we get deeper, since it is misleading. For example, in Fig. 1, in the last layer K23, K_33, and K_32 are in the same graph, while in the upper levels they are not. The whole diagram should just be a stacking of "squares", like below (hope it doesn't get distorted, since it took a while to draw). O----->---------O | | V ^ | | O----->---------O | | V ^ | | O----->---------O | | V ^ | | O----->---------O 2) The authors mention graphs several times in the paper but never really define formally what they mean by it. I am convinced that the whole paper can be written without referring to graphs ever, just sum and composition of sparse operators. However, if the authors do want to mention graphs, you should formally relate the adjacency matrix of these graphs with the sparsity patters of the kernel matrices. The same goes regarding the mentioning of GNNs. I believe the authors can avoid mentioned them at all, and just present the architerure of their NN as is. However, if you do want to mention GNNs, you need to give full details of how your method translates into a GNN, not just say that it does. E.g. from (12) and (13), it is not clear which graph you're using in the GNN. It could either be the metagraph in Fig.1, which I re-expressed above, or a fully detailed graph where, in each level, you express K as a graph operation as well. Please use a latex Definition environment. 3) The authors need to clarify what they mean by "complexity" in different parts of the paper. Is it the complexity to compute K? Is it the complexity to compute K v? Is it the complexity to compute the full map? Also, the authors need to give more detail on how these complexities are computed when more, or less, "tricks" are used. For example, at some point the authors mention that the domain of integration in eq. (3) is truncated to a ball. This corresponds to sparsifying K before doing K v. Is this truncation happening at every level of the the downward and upward pass? I.e. do we use this truncation in all matrix multiplications in eq. (12) and eq. (13)? Also, in addition to this truncation, you also use a Nyström approximation, which corresponds writing K as a product. How exactly does this affect computation time? When explaining these things, always compare what would happen if, e.g. I computed eq. (8) without any "tricks", and if I computed eq. (8) using sparsifying "tricks". Be specific, be rigors. Here it matters, when mentioning Banach spaces it doesn't. 4) At some point there is a disconnect between the flow of ideas that are guiding the development of the heuristic. You start with writing u(x) as solution for eq. (2), fine. Then you say that the r.h.s. of eq. (2) can be approximated by the action of a kernel as in eq. (3). Here I already see a problem as if u(x) = 0 in the r.h.s. of eq. (2) I still get something on the l.h.s. of eq. (2), but if u(x) = 0 in the r.h.s. of eq. (3) I get also zero on the l.h.s. of eq. (3). Then you seem to try to compensate the difference between eq. (3) and eq. (2) by introducing a matrix W. However, I still do not see how W can capture the term with f(x) in eq. (2). I also do not understand why the effect of W cannot be captured by \Kappa_a. Then you extend the dimension of the problem and you introduce activation functions. At this point, since you're learning \Kappa_a and W, you no longer are using any knowledge of the fact that you started from eq. (2). You could have started with eq. (4) directly, and things would still make sense. What's the point of the journey then? Then you explain in eq. (5) how the discretization of the domain affects calculation of eq. (4). Then you introduce the decomposition in eq. (8) as a matrix decomposition, and then clarify that these are actually tensors. Finally, you do not use eq. (8) at all as it is, but you stick an activation function in between every matrix multiplication/sum and offer eq (12) and (13) as your actual algorithm. Here again I see no reason why we need W, as this can be absorbed into K_ll. While this story is going you, you also mention here and there some truncations via B(x,r), graphs, domain discretization, and Nyström sampling, without never being very formal about these things. 5) Many solvers' algorithms are able to guarantee that some nice mathematical properties are kept. For example, that we do not lose mass, or charge, when solving a physics-related continuous PDEs via methods that are inherently discrete. They use symplectic integrators, etc. How does learning F^\dagger behave in this regard? Is it possible to do the training such that some nice conservation properties are kept? Can you illustrate, at least numerically, how well, or not, we conserve certain properties in e.g. Hamiltonian systems?

Correctness: The authors propose an heuristic to solve PDEs. It is based on interesting assumptions, but, as an heuristic, it cannot be said to be correct or incorrect. The authors do show, numerically, that it seem to produce good quality solutions.

Clarity: The language is clear. The level of detail with which different concepts are explained seems kind of random. Sometimes very formal in things that are irrelevant. Other times omitting details in what actually matters. See above.

Relation to Prior Work: They do mention other NN-based methods to solve PDEs, ODEs. It would be good to expand a bit more on current more "classical" methods. Even if it turns out that current methods are actually much faster than what the authors are introducing, the idea of replacing parts of numerical packages with non-linear functions learn from data is interesting.

Reproducibility: Yes

Additional Feedback:


Review 2

Summary and Contributions: The authors propose a generalization of graph neural networks that mimicks the hierarchical construction of multipole approximation. The idea is sound, but the technique is only shown for solving some simple 1D PDEs.

Strengths: The idea is sound, although I suspect limited novelty. There are insufficient benchmarks to demonstrate the value of the approach - I discuss in the following section.

Weaknesses: This looks like many existing works, which have more carefully demonstrated their usefulness: i.e. the MgNet from jinchao xu's group, and superficially this isn't so different from looking at graph convolutions with maxpooling to coarsen hierarchically - I'm curious whether this could simply be implemented as such a network with skip connections and get an equivalent architecture. In any event, the only metrics shown to gauge the stengths of the method are for solving 1D PDE, and the results are remarkably poor (>1% error for all methods considered, including this one). Several ML methods perform better than this (pinns, the deep ritz method, etc), and if one were to use a traditional FMM-based PDE solver one could get orders of magnitude better answers. Of course, this differentiable generalization may be applicable in ML contexts, however the authors structured the paper claiming to impact the solution of PDEs and the results were in my opinion not sufficiently remarkable to merit publication. I will note that I think the research direction pursued here is a solid one and the authors should continue to develop this idea, but hopefully generate some more compelling results for next publication.

Correctness: As far as I can tell yes.

Clarity: Relatively clear. There are some technical details in the first half of the paper where the details of multipole approximations aren't well represented (e.g. "Based on the insight that long-range interaction are smooth, FMM decomposes the kernel matrix into different ranges and hierarchically imposes low rank structures to the long-range components" is not accurate - long-range interactions are generally *not* smooth, and the original FMM was generated for N-body problems specifically where one can exploit regularity. While this is nitpicky - this has consequences later on, where this is used to solve burgers - a hyperbolic problem with shocks and where long range correlations are not smooth)

Relation to Prior Work: The authors mention some similar works but miss the reference I mention previously which do a much better job demonstrating the usefulness/value of these types of hierarchical approaches.

Reproducibility: Yes

Additional Feedback:


Review 3

Summary and Contributions: This paper proposes a new graph neural network model based on the multipole graph neural operator. This model allows a multiscale analysis of graph and thus tackle one of the main drawback of graph neural network. Moreover in order to have a distance for edge, they use kernels thus allow further extension.

Strengths: The main contribution is the multipole operator which permits a multiscale view of the graph. Then the V cycle model shows some similarities with the classical message passing method used for GCN and in fact extend this common model. The multiscale properties of the multipole tackles the one strong issue of the GCN as it permits to involve both local and global part of the graph when learning.

Weaknesses: The behavior of the new model for large complex graph is unclear. The experiment are not tested on classical dataset like Cora and others.

Correctness: The method seems correct.

Clarity: The paper is mostly easy to read. However there is a little confusion on what is really learned in the model, especially on the kernel part.

Relation to Prior Work: The relation to previous work is sufficient.

Reproducibility: Yes

Additional Feedback: There is a little confusion on the learned part. The authors talk about Nymstrom method to deal with kernel, but in the proposed code there is no kernel computation. Please be more precise on these part. Secondly, perhaps the factorized equations would give some insight on the method compared to classical graph neural networks. POST REBUTTAL: Since this work especially focus on PDE, it not ready yet for the GNN community (need comparison against other GNN). Furthermore, since PDE are the main problam to solve, we need comparison against PDE solvers.

[Author Response · NeurIPS 2020]

**Compared to traditional solvers:** Classical methods such as FEM/FDM are designed to solve a PDE once for a given parameter. In contrast, our method approximates the solution operator of a PDE i.e. the mapping from parameter to the solution. The goal of this work is to propose a data-driven surrogate model which offers flexibility in downstream applications as it can produce accurate approximations cheaply and quickly, rather than propose a method for solving PDEs. Our method is consistent in function space i.e. the error is independent of the resolution of the input or output functions which is a crucial property for application that very few deep learning methods posses.

- Traditional solvers: solve one instance of the equation at a time; require the explicit form of the PDE; have a speed-accuracy trade-off based on resolution: accurate on fine grids, less accurate on coarse grids.

- Neural operator: learn a family of equations from data; don't need the explicit knowledge of the equation; much faster to evaluate than any classical method; no training needed for new equations; error rate is consistent with resolution.

**Example: inverse problem of 1-d Burgers' equation:** Suppose we wish to use a function-space MCMC method to find the Bayesian solution of the inverse problem associated with the 1-d Burgers' equation: given a noisy solution at time 1, we wish to sample the distribution of possible initial conditions. This requires tens of thousands of evaluations of the equation with different initial conditions. Using the classical approach with which we generate our data (details in Appendix), each forward run takes around 2 seconds, which amounts to **28 hours** of computation time for 50,000 MCMC steps. Our approach approximates each solution in only 0.01 seconds, amounting to **8 minutes** of computation time. The computational savings are much further embellished when considering more complex PDEs.

**Review 1:** Thank you for the detailed review and helpful suggestions. Methods like those in Chebfun are not designed for parametric equations and are much slower compared to our method; we are happy to add a demonstrative example. Furthermore, conservation properties are kept so long as they are in the data; we are happy to add an example.

- **Technicality:** While we agree that certain sections are more mathematically formal than others, we have done this in order to balance reader comprehension with page-limit requirements. We believe it is important to concretely way out our problem of interest while not so important to give the correct function space every time.

- **Figure:** We appreciate your figure suggestion and would be happy to use it with your permission.

- **Graphs:** While we agree that our methodology can be stated without graphs or GNNs, they are currently the only efficient deep learning tool for implementing it. Furthermore graphs allow for a straight forward extension to domains that are manifolds, a direction we wish to pursue in the future.

- **Complexity:** We measure complexity in terms of the sparsity of $K$ as this is what affects all computations. Originally this complexity is $O(n^2)$. Truncating the integration reduces it to $O(n^2 r^d)$ $(r < 1)$, while the Nyström approximation brings this to $O(m^2 r^d)$ $(m \ll n)$. These approximations are indeed used throughout the V-cycle, a point we will clarify in our revision. (line 140. - 152.)

- **Presentation:** Note that $u(x)$ cannot be identically zero, please see the cited theorem. Furthermore we treat $f(x)$ as fixed and therefore omit it from the architecture. While we agree that the approximation from (2) to (5) is not precise, neither is neural network design. We simply use it as guiding principle, combining it with deep learning intuition such as the addition of the $W$ matrix which makes learning $K$ easier.

**Review 2:** Thank you for your comments. Please note that our work considers both 1-d and 2-d problems (Section 4).

- **MgNet:** While MgNet is an interesting work, it is substantially different from ours. In particular, they employ the multigrid method which is based on a residual correction while we use the multipole method which is based on a decomposition of the integral kernel. Furthermore our graph-based convolutions allow our method to achieve consistent error for any discretization while continuous convolutions are tied to a particular grid due to their local parameterization. Lastly, MgNet is only applied to image classification tasks (Cifar) while we do function to function regression, arguably, a much more general task. We would be happy to add a citation and will try to include a numerical comparison in the final version, but unfortunately their code is currently not online.

- **Other methods:** While methods such as PINNs and Deep ritz are interesting, they are not suitable for our goals. They both require solving a non-convex optimization problem for each parameter of interest which is slow and costly (weeks of computation for the above inverse problem). Such methods are useful when the underlying physical domain is high dimensional and FEM becomes infeasible, but this is not a setting our work considers. In the parametric setting, we are not aware of any methods with better accuracy than ours.

**Review 3:** Thank you for your nice suggestions. We designed our method to approximate parametric PDEs and therefore never tested it on standard graph datasets such as Cora. While this is an interesting future direction as quickly processing global information on large graphs is crucial in applications, it is well beyond the scope of the current work.

[Meta-Review · NeurIPS 2020]

I agree with the authors that R1's concerns are not relevant to the acceptance decision and have removed their review from consideration. R2 raised the concern that there are insufficient benchmarks to judge the value of the work; the author rebuttal countered that the baselines identified by R2 do not attack the same use case as the proposed algorithm. I concur with this assessment. R2 also pointed out that the method was evaluated on 1d problems only; the authors rebutted that the method was demonstrated on both 1- and 2d problems, and gave an example of a Bayesian inverse problem that motivates this method even in low dimensions. R4 recommended accept because of the novelty of the proposed multipole graph neural operator. Taking the reviews and author rebuttals into consideration (excluding R1), while I see value in this work, the conclusion is that the potential impact of the method is not clear. What is encompassed in the class of problems for which this heuristic will work: will it be limited to Bayesian solutions of 1d inverse problems, as the example offered by the authors in the rebuttal? Unfortunately, NeurIPS is a selective conference, and both novelty and impact must be clear. The former is evident, but because the latter is unclear, I reject. SAC metareview: The paper received 3 reviews, which were not reliable enough, so we asked two additional reviews, edited below. This triggered additional discussions. We decided this was a borderline accept. We trust the authors will take into account the several remarks we were able to gather and **update** their camera-ready. Reveiewer A: I think the experiments are lacking in terms of variety, more concretely the proposed architecture is only compared to others in a 1-D problem. It also feels a bit weird that they did evaluate on a 2-D equation but didn't report comparisons with other methods there (I didn't have access to the appendix). Having said that, I think this paper is relevant and important in the area of deep learning to learn solutions from parameters This work combines essentially 3 ideas: - (1) using GNNs for learning PDEs from data; relevant work w.r.t. this is cited to the best of my knowledge - (2) expressing the action of the integral operator with graph neural network message passing with a particular kernel function, which allows seamless multi-resolution. Some interesting works by Welling's group feel very relevant here: Gauge Equivariant Mesh CNNs Anisotropic convolutions on gemoetric graphs (de Haan, Weiler et al.), Gauge Equivariant Convolutional Networks and the Icosahedral CNN (Cohen et al.). There is non-zero probability that I misunderstood something in either this work or the two Welling works and they are not relevant. However, if they are, they also make me think that the application of their ideas is not as trivial for non-Euclidean meshes. - (3) hierarchical message passing [some work is cited, but two relevant works in physics-based problems aren't: Flexible Neural Representation for Physics Prediction (Mrowca et al.), Learning particle dynamics for manipulating rigid bodies, deformable objects, and fluids (Li et al.). Note that both works deal with particle-based simulations of objects, not PDEs in continuous spaces. - Even with the non-cited work I still feel this work is novel and non-trivial. Moreover, the combination of these three ideas is of particular importance and thus this paper still provides an important stepping stone for the community. The paper is mostly sound and well explained. - As I mentioned in related work, some of Welling's work shows that putting a graph on a mesh should be done with care because traditional GNNs impose that all edges are created equal, but this is too restrictive for most meshes. Instead GNNs on meshes should satisfy a set of equations depending on the local tangent space, as well as satisfying some global symmetries. I suspect (but can't say for certain) that this may cause trouble for the approach presented in non-Euclidean manifolds (say, a sphere) The contribution is important and a good one. It is more relevant to a generic audience than most papers, IMO. Moreover, both the GNN and the Deep Learning for Science communities are growing. Overall thoughts: - I think the combination of ideas and the application is relevant, important and pretty well implemented. My main concern is that the experiments were only done in a 1-D equation and the 2-D equation only reported times and did not compare with alternatives. Moreover, the proposed approach should be able to run on non-Euclidean manifolds and I have doubts (though I'm not certain) that some issues may arise there, as described in technical soundness. If there were convincing comparisons in a 2-D setting I would argue for a soft accept and if there were satisfying experiments in a non-Euclidean manifold with good explanations on how to adapt things there I would argue for a strong accept. Reviewer A update, after being given access to the appendix: Thanks for sending the appendix. Looking at it carefully, I agree they tested 2D Darcy flow against baselines, they probably didn't show them in the main text for the reason I suspected: because multiple baselines beat their model. On a related note, thinking about these values more, the premise on increasing mesh size for the sake of doing so is erroneous IMO: if a lower mesh size gives a lower error, then just go with it and you'll also have a cheaper cost. This is relevant for the GCN baseline which they say works well on an s=16, which they don't show as well as the FCN baseline, which gets great results (better than the proposed method) at low resolution. MGKN should also aim at getting better performance, not just equally good, at higher resolutions. Unless I misunderstood something, for each baseline and each mesh size s they should have reported the min for all s'<=s (technically, the test after cross-validation to find best s'). This understanding would also experimentally contradict the need for denser meshes, which is the main premise of the paper. I do think denser meshes are valuable in principle, which is why I like the idea behind this paper, I'm just doubtful the experiments are showing this need. Reviewer B: The method of the paper is definitely novel and, I think, very much deserves to be published. The authors are correct in their reply to reviewer 2, when they say that the multigrid method and the FMM are essentially unrelated (except very superficially by the presence of grids or varying coarseness).[...] For the same reason, the authors' method and MgNet are essentially unrelated. The authors' use of the term "V-cycle" might be partially to blame for this confusion, since that term is used frequently in the multigrid community but is rarely used in the FMM community (though its meaning in that context is clear) [...] Weaknesses: 1. I agree with reviewer 2 that, for hyperbolic problems, the theory behind the method is much less clear. On the other hand, the method's applicability to elliptic PDEs whose solutions depend in a nonlinear way on a parameter (equation (14), for example) is compelling. 2. Figure 1 is misleading, since it shows a nearest neighbor graph (with O(N) interactions). The lowest level should be have every node connected to every node (O(N^2) interactions), the next level should have a smaller subset of those nodes, all once again connected to one another, and so on. Their method uses the lowest level to compute nearby interactions, the next level to compute more well-separated interactions, the level above that to compute even more well-separated interactions, etc. They use NNs to construct the kernels on each level, as well as the kernels of the "transition" matrices which map between different levels.